# Comparison of Nicotine Dependence and Biomarker Levels among Traditional Cigarette, Heat-Not-Burn Cigarette, and Liquid E-Cigarette Users: Results from the Think Study

**DOI:** 10.3390/ijerph18094777

**Published:** 2021-04-29

**Authors:** Guillaume Rudasingwa, Yeonjin Kim, Cheolmin Lee, Jeomkyu Lee, Seunghyun Kim, Sungroul Kim

**Affiliations:** 1Integrated Research Center of Risk Assessment, Soonchunhyang University, Soonchunhyang-Ro 22, Asan 31538, Korea; guillaumer1992@gmail.com; 2Department of ICT Environmental Health System, Graduate School, Soonchunhyang University, Soonchunhyang-Ro 22, Asan 31538, Korea; duswls0911@naver.com; 3Department of Family Medicine, Healthcare System Gangnam Center, Seoul National University Hospital, Seoul 06236, Korea; bigbangx@snuh.org; 4Division of Respiratory and Allergy Disease Research, Department of Chronic Disease Convergence Research, National Institute of Health (NIH), Korea Disease Control and Prevention Agency (KDCA), Osong 28159, Korea; nihdot@korea.kr (J.L.); shkims00@korea.kr (S.K.)

**Keywords:** nicotine dependence, biomarker, smoking, electronic cigarette, heat-not-burn cigarette

## Abstract

This study aimed to compare Korean smokers’ smoking-related biomarker levels by tobacco product type, including heat-not-burn cigarettes (HNBC), liquid e-cigarettes (EC), and traditional cigarettes (TC). Nicotine dependence levels were evaluated in Korean adult study participants including TC-, EC-, HNBC-only users and nonsmokers (*n* = 1586) from March 2019 to July 2019 in Seoul and Cheonan/Asan South Korea using the Fagerström Test Score. Additionally, urine samples (*n* = 832) were collected for the measurement of urinary nicotine, cotinine, OH-cotinine, NNAL(4-(methylnitrosamino)-1-(3-pyridyl)-1-butanol), CYMA(N-acetyl-S-(2-cyanoehtyl)-L-cysteine), or CEMA (2-cyanoethylmercapturic acid) using LC–MS/MS. The median(interquartile range) nicotine dependence level was not different among the three types of smokers, being 3.0 (2.0–5.0) for TC- (*n* = 726), 3.0 (1.0–4.0) for EC- (*n* = 316), and 3.0 (2.0–4.0) for HNBC- (*n* = 377) only users. HNBC-only users presented similar biomarker levels compared to TC-only users, except for NNAL (HNBC: 14.5 (4.0–58.8) pg/mL, TC: 32.0 (4.0–69.6) pg/mL; *p* = 0.0106) and CEMA (HNBC: 60.4 (10.0–232.0) ng/mL, TC: 166.1 (25.3–532.1) ng/mL; *p* = 0.0007). TC and HNBC users showed increased urinary cotinine levels as early as the time after the first smoke of the day. EC users’ biomarker levels were possibly lower than TC or HNBC users’ but higher than those of non-smokers.

## 1. Introduction

Cigarette smoking remains a major health concern worldwide. Up to 80–90% of lung cancers are attributable to smoking, which is also causally linked to oral, esophageal pharyngeal, renal, and cervical cancers, stroke, coronary heart diseases, and various chronic conditions [1,2,3]. In South Korea, approximately 37% of Korean adult males and 5.2% of Korean adult females were found to be smokers in 2017 [4].

Recently, various electronic nicotine delivery systems (ENDS), including liquid e-cigarettes (EC) and heat-not-burn cigarettes (HNBC), have been widely used as alternatives to traditional cigarettes (TC). HNBC, which is a type of cigarette heated with an electric blade at 350 °C, gives the smoker the true taste of tobacco, with reduced smoke and smell. The preference of smokers has been shifting from TC to EC and HNBC. This change in smoking patterns raises major issues such as the biochemical harmfulness of ENDS to humans, as warned by global health organizations, including the CDC and the World Health Organization (WHO) [5,6]. Since the introduction of IQOS, a type of HNBC, in South Korea in June 2017, such HNBCs have gained popularity, accounting for 10.5% of all total tobacco sales in 2019 [7,8], and were advertised as a product with reduced harm [9]. However, the Korean Ministry of Food and Drug Safety claimed that they may not be as safe as the companies claim [10]. In Korea, according to the third smoker panel follow-up survey based on the Korea National Health and Nutrition Examination Survey (KNHANES 2017), the rate of EC usage reached a peak after the increase in cigarette prices in 2015 but decreased slightly in 2016 [11].

HNBC, EC, and TC mostly contain nicotine and other chemicals, which are inhaled through the mouth [12]. Among daily smokers, smoking patterns tend to maintain relatively stable levels of nicotine in the body, which leads to frequent smoking [13]. Intermittent smokers (ITS) or non-daily smokers also show some evidence of nicotine dependence and difficulty in smoking cessation, thus making them suffer from the adverse effects of smoking [14,15,16]. In fact, nicotine dependence is the determinant of cigarette smoking, because individuals smoke frequently to maintain nicotine levels, thus reducing the possibility of quitting [17].

According to the WHO, the efficacy of EC for quitting smoking has not been systematically evaluated. The basis for the effectiveness of EC for quitting smoking is extremely limited, and conclusions cannot be reached [18]. In a randomized control study conducted in New Zealand in 2013, the 6-month smoking cessation rate for groups using nicotine-containing EC and nicotine patches was 7.3% and 5.8%, respectively, but this difference was not statistically significant. Due to criticisms of the research design, there are still debates over the anti-smoking effects of EC [19]. According to the 2016 Cochran Review, EC, containing nicotine, was associated with a higher smoking rate than EC without nicotine. However, more research is needed on whether EC is superior to other supplements such as nicotine patches [20].

Nicotine dependence complexity has been reported, and multiple instruments have been adopted to evaluate the extent of nicotine dependence [21,22,23]. The Fagerström Test for Nicotine Dependence, which is a standard tool designed for evaluating physical addiction to nicotine, includes questions regarding the time of the first cigarette, the number of cigarettes smoked per day, the difficulty to refrain from smoking, the type of cigarette for which someone would find it difficult to give up on, and the daily smoking frequency [24,25].

One of the effective methods for measuring smokers’ exposure to cigarettes is measuring biomarker levels, which estimates the integrated exposure over a period of time [26,27]. Biomarkers also play an important role when reviewing the potential health risks of tobacco products as well as for evaluating and identifying potential product standards of components in tobacco products [28,29]. Reliable and accurate measurements of tobacco products in human biospecimen are necessary to identify and confirm tobacco use patterns and to assess their potential biological effects in the human population [30]. Exposure to nicotine, cotinine, NNAL, and alkaloids is related to the exposure to nicotine or toxicants present in cigarettes and other tobacco products [31].

Considering the possibility of the increasing interest in new tobacco products, there is a need to lay the foundation for health policy development by identifying user behaviors by cigarette type and surveying exposure levels. Considering the international cigarette consumption rate by cigarette type—TC, EC, and HNBC—comparison of the levels of smoking-specific biomarkers depending on cigarette type or addiction level can be essential to evaluate the relationship between tobacco smoking habits and potential health outcomes. However, study outcomes that contain such information are highly limited. The purpose of this study was to compare Korean smokers’ biomarker levels by type of tobacco product consumed, i.e., HNBC, EC, or TC.

## 2. Materials and Methods

### 2.1. Study Participants and Design

Under a cross-sectional design, the THINK (Tobacco and Health IN Korea) study comprising 3004 adult study participants, aged 19 years or older, including non-smokers and single, dual, or triple users of cigarettes, EC, and cigarette-type EC, were recruited using a convenience sampling method from March to June 2019 (Figure 1). In this study, we excluded participants under 19 or over 65 years of age. Pregnant women, nicotine supplement users, and patients with severe lung disease were also excluded from the selection process. Research subjects (adult male and female smokers) who voluntarily applied through public information advertisements or handouts were randomly selected from hospital outpatient centers or public places, train stations, markets, etc. To analyze urinary biomarkers, urine samples were collected from those (*n* = 832) who agreed to provide their urine.

### 2.2. Questionnaire Development for Investigating the Tobacco Usage Behavior

To conduct the THINK study, we developed a questionnaire asking smokers’ smoking behavior based on the US PATH study questionnaire [32], KNHANES smoker panel third follow-up 2018 [11], and International Tobacco Control survey [33]. Using screening questionnaires, before conducting the main survey, smokers were classified into eight types according to the combination of tobacco products they used, that is, non-smokers/past smokers; TC-, EC-, HNBC-only users; TC–EC dual users; TC–HNBC dual users; EC–HNBC dual users; TC–EC–HNBC triple users. Questionnaires contained questions about the lifetime smoking period, cigarette brand, daily smoking amount (unit of pack was used for TC and HNBC users, and unit of puff per usage and frequency of usage were used for EC for its estimation), Fagerström dependence level, and intention to quit smoking. This study used pictures of tobacco products to reduce classification errors or confusion about the types of new cigarettes. In addition, tobacco product users (TC, EC, and HNBC) were further defined as follows: Current User: In case of responding: ”I use cigarettes every day or sometimes”Past Users: In the case of responding: ”I used it in the past, but not currently” andNon-user: No experience of smoking at all.

At the baseline, we excluded the users of these three products involved in various smoking cessation service programs (visiting smoking cessation clinics, using smoking cessation consultation calls, etc.).

Our survey was conducted using an online survey method (CAWI: computer-aided web intervention) and the help of trained agents of Gallop-Korea, field managers of the Integrated Research Center of Soonchunhyang University, or trained field managers in hospital-based health examination centers (Seoul National University Hospital Healthcare System Gangnam Center; Seoul National University Hospital in Bundang; Soonchunhyang University Hospital in Cheonan; Dankook University Hospital in Cheonan, South Korea). The original study procedures relating to human participants and informed consent were approved by the research ethics committee of Soonchunhyang University (IRB No. 2018-BR-046-03), and written consent was obtained.

### 2.3. Component of the Survey Questionnaires

The survey of the 3004 participants (smokers or non-smokers living with smokers) included questions about age, gender, education, smoking habits (smoking period, smoking volume, nicotine dependence, intention to quit smoking, smoking place, dual use status, and products used). Detailed questions about smoking/non-smoking, smoking habits, and nicotine addiction levels were also included.

Nicotine dependence was measured using the Fagerström score. The Fagerström score was divided into low dependence for a score of 1 to 2 points, upper low dependence for a score of 3 to 4 points, medium dependence for a score of 5 to 7 points, and high dependence for a score of 8 points or more by using the answers for the questions about the waiting time to the first smoke after waking up, controlling the desire to smoke, daily smoking rate, etc.

Because the daily smoking amount could not be calculated for EC users using the parameter “number of cigarette,” unlike for TC or HNBC users, we converted the number of liquid inhalations to estimate the smoking volume and calculated the Fagerström nicotine dependence score according to ISO3308:2012(E): 10 inhalations of liquid = 1 cigarette).

Questions about the experience of exposure to secondhand smoke were also included. Among the eight types of smokers who provided biospecimens (*n* = 832), for those who agreed, a topography assessment using CReSS was conducted. A reward card ($100) was given to research subjects who provided biospecimens and conducted the topography test.

### 2.4. Measurement of Biomarkers

#### 2.4.1. Selection of Biomarkers

In this study, six types of smoking-related biomarkers were analyzed: urine nicotine, cotinine, OH-cotinine, NNAL (metabolite of NNK), CYMA and CEMA (metabolites of volatile organic compounds that is, acrylonitrile and acrolein). These biomarkers were recently selected as major smoking-related substances [34].

#### 2.4.2. Analysis of Biomarkers

Biomarkers were analyzed using liquid chromatography and tandem mass spectrometry (LC–MS/MS) [30,35,36]. During our sample preparation period, 0.1 mL of urine was diluted 10 times with an acetonitrile and water (50:50 *v/v*) mixture for cotinine and OH-cotinine, nicotine, NNAL, CEMA, and CYMA analyses. The water blank was replaced with 0.1 mL of HPLC-grade water instead of urine. Then, 100 μL of the internal standard mixture (10 µg/mL) was added, and the samples were centrifuged at 5 °C for 30 min at 3600 rpm.

After centrifugation, urine samples were passed through a filter of 0.2 µm as pore size, and the supernatant solution was put into a vial; 20 μL was injected into the LC–MS/MS. For the separation of cotinine, OH-cotinine, and NNAL, the Waters Acuity 1.8 μm column was used. Low concentrations of NNAL were selected and analyzed by liquid–liquid extraction.

Our selective analysis was based on the following: cotinine: 177.1; Cotinine-d3: 180.1; OH-cotinine: 192.6; OH-cotinine-d3: 196.1; NNAL: 210.0; d3-NNAL: 213.0 *m*/*z*; CEMA: 236.0; and ^13^C3-CEMA: 239.0 CYMA: 217.0; and d3-CYMA: 220.0 were used as precursors. Cotinine: 80.1; Cotinine-d3: 80.0; OH-cotinine: 80.0; OH-cotinine-d3: 80.0; NNAL: 92.9; d3-NNAL: 93.0 *m*/*z*; CEMA: 105.0; ^13^C3-CEMA: 105.0; CYMA: 158.0; d3-CYMA: 176.0 *m*/*z* were used as products.

The detection limits in this study were as follows: 0.15 ng/mL (cotinine, OH-cotinine, nicotine), 0.5 ng/mL (CYMA, CEMA), and 1.0 pg/mL (NNAL). In this study, the quantitation limit was selected as 0.3 ng/mL (cotinine, OH-cotinine, nicotine), 1.0 ng/mL (CYMA, CEMA), and 4.0 pg/mL (NNAL).

#### 2.4.3. Urine Sample Collection

Spot urine samples were collected from volunteers who provided answers to the survey questionnaire and agreed to provide urine. Urine samples were collected at the hospital’s examination center, Soonchunhyang Cheonan hospital smoking cessation support center, and Soonchunhyang University’s risk assessment center and Seoul metro station. We collected 30 mL or as much urine as provided in a conical centrifuge tube (50 mL).

The field coordinators stored the sample tubes in an ice box and kept it until it arrived at Soonchunhyang University’s analytical lab. After checking study volunteers’ IDs, samples were delivered to the laboratory and put in a freezer at (−20 °C) until the analysis. Field inspectors were responsible for accurate labeling, sampling, and storage on cards, envelopes, and tubes, re-checking labeling and transporting to the risk assessment center at Soonchunhyang University for analysis. All analytical procedures were conducted by blind testing in accordance with the internal quality control protocol of the risk assessment institute of Soonchunhyang University.

### 2.5. Data Analysis

Descriptive statistics (frequencies, percentages, means, medians, and interquartile ranges) were used to describe the study population and smoking behaviors. Differences in demographics and smoking-related characteristics by sex were determined using Fisher’s exact test and the Mann–Whitney *U* test (Wilcoxon rank-sum test). Statistical significance was defined as a *p* value < 0.05.

A log-transformed linear regression analysis of outcome variables was conducted to account for the right-skewed distribution. The regression coefficients were back-transformed to estimate the geometric mean ratios when comparing different levels of the independent variable. Additionally, we conducted a sensitivity analysis among a subset of the population with low puff counts. As we observed, there were no substantial differences in the findings; the full population is reported herein. Variables that were significant (or borderline significant) in the univariate models were included in the multivariate regression model. All analyses were conducted in SAS version 9.1 (SAS Institute, Cary, NC, USA).

## 3. Study Results

### 3.1. Demographic Characteristics of the Study Subjects

The demographic characteristics of a total of 1586 study participants are summarized in Table 1. For each cigarette type, male participants were the majority, and more than about 70% were those aged 30 or older. Most of our study participants were university graduates. The proportion of participants with a high income level ($5000/month) was larger among new-type cigarette-only users (EC: 54%; HNBC: 58.4%) compared to traditional cigarette users (TC: 46%).

As shown in Table 1, the proportion of the number of cigarettes smoked per day differed within smoking groups (*p* < 0.05). According to the Fagerström index, 4% or 2% of TC or HNBC smokers, respectively, were considered to be highly nicotine-dependent (score ≥ 4).

### 3.2. Comparing the Concentration Levels of Biomarkers by Tobacco Product Type

Among those who provided urine samples, the concentration of biomarkers by single cigarette-type users was as follows: for liquid-type EC-only users, the levels of NNAL (8.3 pg/L), cotinine (322 ng/mL), and CEMA (12 ng/mL) were significantly lower than those of regular TC-only users (32 pg/mL, 730 ng/mL, and 166 ng/mL, respectively) (Table 2).

We observed that the concentrations of biomarkers for HNBC-only users were not statistically different, except for NNAL (*p* = 0.0106) and CEMA (*p* = 0.0007), compared to those for TC-only users. HNBC-only users’ NNAL and CEMA concentrations were 14.5 pg/mL and 60.5 ng/mL, respectively, and were statistically lower than the levels found in TC users (32 pg/mL and 166 ng/mL, respectively). Regardless of the cigarette type, the biomarker levels of single users were statistically different from the levels of non-smokers.

In the case of TC and EC or TC and HNBC dual users, the biomarker levels were similar to those of TC-only users. In this study, we did not include further interpretation of the level of biomarkers of EC and HNBC dual users due to their small sample size (*n* = 6).

### 3.3. Relationship between Nicotine Addiction Level and Cotinine

In this study, the nicotine addiction level was evaluated using cotinine, one of the most common smoking-related biomarkers. The table below shows the concentration of cotinine by the level of waiting time to the first smoke after waking up (31 min or more, 6–30 min, 5 min or less).

As shown in Table 3, we found that in most cases, except for EC-only users, the shorter the waiting time to the first smoke after waking up (within 5 min, 6–30 min, 31 min or more), the higher the concentrations of cotinine observed. In the case of the EC-only users, the median value of urinary cotinine was not different between those who smoked their first cigarette within 5 min after waking up and those who smoked within 6–30 min after waking up. Due to the relatively small number of samples, we did not conduct further interpretation on this.

Similarly, using the Fagerström nicotine addiction index, the association between the addiction levels and the urinary cotinine levels was investigated (Table 4). Since it is difficult to define the number of cigarettes smoked by EC users, as defined in the ISO smoking test, we defined 10 inhalations as one cigarette [37]. Using a survey-based daily inhalation frequency and the definition above, we calculated the total smoking amount per day.

In the case of TC-only and HNBC-only users, we observed that urinary cotinine levels increased as the Fagerström index score increased. In the case of EC-only users, the number of samples was small; therefore, it is judged that additional interpretation is meaningless. Since dual users were not asked questions for deriving the Fagerström index, we did not include these results.

### 3.4. Relationship between Cigarette Smoke Intake and Nicotine Addiction Levels Adjusted for Urinary Cotinine Level

Using the levels of biomarkers among single-type cigarette smokers, we estimated the association of urinary biomarker levels with the daily cigarette smoking amount (Appendix A). As expected, we found that there was a dose–response relationship between urinary cotinine levels and daily smoking amounts (6–10 cigarettes followed by 11–15 cigarettes, compared to 5 or less cigarettes), among TC (*p* < 0.05) or HNBC (*p* < 0.05) users after adjusting for age and sex. We found a similar dose–response relationship among EC users, but it was not statistically significant.

We further analyzed our data to evaluate the association of daily smoking amount and Fagerström nicotine addiction determinant factors, adjusted for age, sex, and urinary cotinine levels (Table 5). According to our study results, smokers who smoke their first cigarette within less than 5 min after waking up presented the largest daily smoking amount. Other Fagerström addiction determinant variables showed slightly different results depending on the cigarette type. Urinary cotinine levels were positively associated with the daily smoking amount after adjusting for the Fagerström nicotine addiction determinant factors, age, and sex.

## 4. Discussion

### 4.1. Biomarkers Levels by Cigarette Type

In this study, we found that the median values of urinary cotinine, OH-cotinine, and nicotine among TC-users were 730, 2227, and 1121 ng/mL, respectively, and the median values of CYMA and CEMA were 180 and 166 ng/m, respectively. The median concentration of NNAL was 32 pg/mL.

The concentrations of biomarkers (cotinine, OH-cotinine, nicotine, etc.) for HNBC-only users were at statistically similar levels as those of TC-only users. These results conform with those of another study that determined the concentrations of all TSNAs (including NNAL) and CEMA among HNBC users [34].

In the case of EC-only users, all biomarker levels were lower. The findings of our study are in accordance with a cross-sectional observational study conducted in the U.S. finding lower concentrations of total NNAL, CEMA, and other biomarkers among EC users [38].

### 4.2. Association of Biomarker Levels with Nicotine Dependence Levels

In this study, among TC users and HNBC-only users, a statistically positive association between addiction level and daily smoking amount was observed, where the higher the level of nicotine dependence, the shorter the time of the first smoke after waking up (within 5 min, 6–30 min). Various studies have found an association between higher cotinine levels and a shorter time to smoking cigarettes after waking, which might mainly be due in part to more intense smoking in response to overnight abstinence and can predict cancer [39,40].

For TC users, the concentration of urinary cotinine was significantly influenced by the first cigarette smoked after waking up, according to the evaluation scales of Fagerström nicotine dependence, adjusting for age, gender, and household income levels. Cotinine concentrations in groups that smoke within 5 min and those that smoke within 6–30 min showed dose–response relationships (*p* < 0.05) (Table 5). HNBC also showed a positive dose–response relationship, but it was not statistically significant.

In the case of TC- or HNBC-only users, the association with daily smoking amount was found to increase with the urinary cotinine concentration as the amount of smoking increased, compared to that in the group that smoked less than 5 cigarettes per day, and this association was statistically significant. For all three types of tobacco users, the higher the concentration of cotinine in the urine, the higher the daily smoking volume, and this association was shown to be statistically significant.

However, due to the relatively small sample size of EC users, the comparison of biomarker levels among EC users was limited, and the interpretation of the results of the EC users should be conducted with care. Differently from TC or HNBC, we considered that 10 puffs of the liquid was equal to one cigarette. It is necessary to conduct further analysis in the future to validate our findings from EC users.

Smoking cessation treatment requires an accurate understanding of the dependence on nicotine [41]. The levels of nicotine, cotinine, or NNAL (a tobacco-specific carcinogen) in biospecimens act as an indicator of the amount of tobacco smoked [42] and is one of the best indicators of potential health effects. The period to smoking the first cigarette after waking up has become increasingly recognized because it is also correlated with many other forms of dependency, including cessation of smoking [41] and tolerance [43]. The Fagerström nicotine dependence score was useful, but we needed some modification to estimate the smoking amount for EC users. Because we conducted this study with a cross-sectional design, we cannot generalize our study results to the Korean population; however, our study provides valuable information on smoking-related biomarker levels of EC and HNBC users compared to TC users. The authors believe that stakeholders can refer to our study results for comparison purposes if needed.

## 5. Conclusions

The nicotine dependence level among users of TC, EC, and HNBC was not significantly different. However, a positive dose–response relationship between urinary cotinine levels and daily smoking amount was found among TC or HNBC users. The time to smoking the first cigarette, one of key Fagerström nicotine addiction index questions, was significantly associated with the daily smoking amount.

## Figures and Tables

**Figure 1 ijerph-18-04777-f001:**
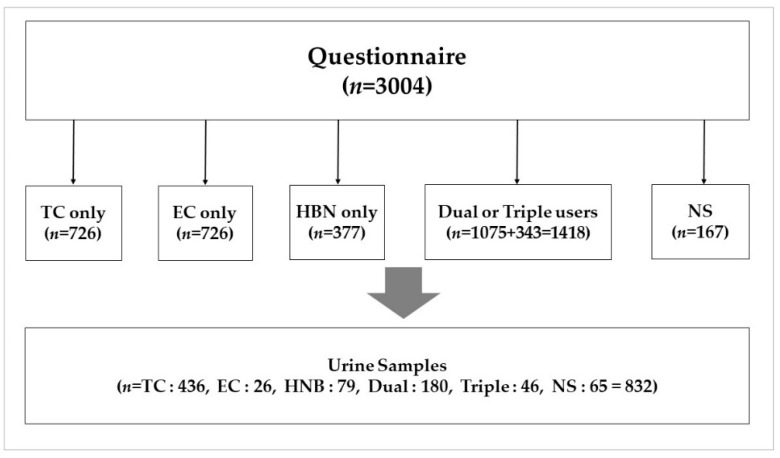
Study framework for smoking behavior including nicotine dependence level and biomarker levels among study participants (among nonsmokers or single-type cigarette users, those who provided a completed questionnaire and urine samples were 63 for NS and 403, 24, and 76 for TC, EC, and HNBC, respectively.).

**Table 1 ijerph-18-04777-t001:** Proportion of study subjects according to age, household income, education level, residential area, marital status, smoking status, and Fagerström nicotine addiction level.

		Nonsmokers (*n* = 167)			TC Only (*n* = 726)			EC Only (*n* = 316)			HNBC Only (*n* = 377)		
		N	%	*p*-Value	N	%	*p*-Value	N	%	*p*-Value	N	%	*p*-Value
Gender	Male	56	33.5	<0.0001	621	85.5	<0.0001	229	72.5	<0.0001	300	79.6	<0.0001
	Female	111	66.5		105	14.5		87	27.5		77	20.4	
Age group (Years)	–29	37	22.2	<0.0001	182	25.1	<0.0001	91	28.8	<0.0001	35	9.3	<0.0001
	30–39	60	35.9		135	18.6		108	34.2		106	28.1	
	40–49	41	24.6		176	24.2		73	23.1		147	39.0	
	50–59	20	12.0		159	21.9		42	13.3		85	22.6	
	60–69	7	4.2		65	9.0		2	0.6		3	0.8	
	70 or older	2	1.2		9	1.2					1	0.3	
Education	Univ	146	87.4	<0.0001	523	72.0	<0.0001	251	79.4	<0.0001	336	89.1	<0.0001
	High school	21	12.6		203	28.0		65	20.6		41	10.9	
House income (M, Won)	5+	93	56.3	0.154	336	46.3	0.123	170	53.8	0.122	220	58.4	0.101
	4.99 or less	74	43.7		390	53.7		146	46.2		157	41.6	
Marriage status	Married	102	61.1	0.076	378	52.1	0.397	150	47.5	0.527	261	69.2	0.076
	Others	65	38.9		348	47.9		166	52.5		116	30.8	
Fagerstrom score	1	NA	NA	NA	283	38.9	<0.0001	140	44.3	0.017	151	40.0	<0.0001
	2				224	30.9		116	36.7		135	35.9	
	3				191	26.3		58	18.4		83	22.0	
	4				28	3.9		2	0.6		8	2.1	
Cigarettes per day	–5	NA	NA	NA	135	18.6	<0.0001	181	57.3	0.0312	83	22.0	<0.0001
	6–10				210	28.9		48	15.2		125	33.2	
	11–15				167	23.0		25	7.9		91	24.1	
	16–20				162	22.3		20	6.3		62	16.5	
	21–25				19	2.6		7	2.2		8	2.1	
	26–30				24	3.3		8	2.5		5	1.3	
	31+				9	1.3		27	8.6		3	0.8	

**Table 2 ijerph-18-04777-t002:** Comparison between tobacco product type and biomarker concentration levels among subjects.

(Unit: ng/mL, Except NNAL: pg/mL)				
		Nonsmokers or Single Users		
		Median	P25	P75
Non-smokers	NNAL	4.9	4.9	4.9
(*n* = 63)	CYMA	0.4	0.4	304.7
	COT	0.9	0.9	0.9
	OHCOT	2.6	2.6	2.6
	NIC	3.9	3.9	149.5
	CEMA	17.5	10.0	95.6
TC only	NNAL	32.0	4.9	69.8
(*n* = 403)	CYMA	179.9	0.4	592.4
	COT	729.5	185.8	1342.6
	OHCOT	2227.1	500.3	4802.3
	NIC	1121.1	42.3	4558.7
	CEMA	166.1	25.3	532.1
EC only	NNAL	8.3	4.9	25.4
(*n* = 24)	CYMA	0.4	0.4	257.3
	COT	322.2	0.9	722.8
	OHCOT	820.3	172.1	2714.2
	NIC	339.1	3.9	4473.6
	CEMA	11.9	10.0	92.7
HNBC only	NNAL	14.5	4.9	58.8
(*n* = 76)	CYMA	206.2	0.4	526.2
	COT	765.5	305.9	1511.3
	OHCOT	2824.7	848.0	7108.8
	NIC	874.5	51.4	7446.3
	CEMA	60.4	10.0	232.0

Note: Detection limit: 0.15 ng/mL (cotinine, OH-cotinine, nicotine), 0.5 ng/mL (CYMA, CEMA), 1.0 pg/mL (NNL).Results were included for those who both completed the questionnaire and provided urine samples (Nonsmokers: 63, TC-only smokers: 403, EC-only smokers: 24, HNBC-only smokers: 76). *p*-values from Wilcoxon rank-sum test (TC vs. HNBC) were 0.0106, 0.6436, 0.4517, 0.0919, 0.7734 and 0.0007; The values between TC and EC were 0.0048, 0.0524, 0.0168, 0.0858, 0.3199 and 0.0004 for NNAL, CYMA, Cotinine, OH-cotinine, Nicotine and CEMA, respectively.

**Table 3 ijerph-18-04777-t003:** Median and interquartile ranges of urinary cotinine level by waiting time to the first smoking after wake-up.

	Waiting Time to First Smoking after Wake-Up (minutes)	Cotinine (ng/mL)			
		N *	Median	P25	P75
TC	31+	191	360.0	0.9	981.8
	6–30	122	861.5	362.2	1523.5
	–5	87	1271.5	707.2	1834.0
EC	31+	16	16.8	0.9	458.5
	6–30	5	755.7	425.4	1866.7
	–5	3	632.8	298.2	724.8
HNB	31+	44	614.4	326.4	1305.6
	6–30	23	844.2	405.3	1394.3
	–5	8	1873.4	227.0	3239.8

* Excluding those who did not respond to the questionnaire or who did not agree with the CO measurement.

**Table 4 ijerph-18-04777-t004:** Frequency of cigarette smoking evaluated by the Fagerström test for nicotine dependence.

	Fagerström Index *	Cotinine			
		N **	Median	P25	P75
TC	1	149	256.9	0.9	778.2
	2	113	932.9	448.7	1562.2
	3	119	985.9	439.1	1638.3
	4	18	1195.1	527.0	1831.8
EC	1	11	0.9	0.9	32.8
	2	9	573.2	310.6	1866.7
	3	3	724.8	632.8	755.7
	4	0			
HNB	1	33	438.1	91.1	861.3
	2	25	778.6	350.9	1798.3
	3	14	1491.1	844.2	2299.8
	4	3	1679.5	998.6	3067.5

* Fagerström Index: 1, low dependence; 2, low medium dependence; 3, medium dependence; 4 high dependence. ** Excluding those who did not respond to the questionnaire or who did not agree with CO measurement.

**Table 5 ijerph-18-04777-t005:** Association between daily smoking amount and nicotine addiction levels adjusted for age sex and urinary cotinine level.

		TC (AIC = 2175)		EC (AIC = 330)		HNB (AIC = 403)	
		B	*p*-Value	B	*p*-Value	B	*p*-Value
Time to first cigarette (Ref: 61+ min)							
	–5	0.254	<0.0001	1.675	<0.0001	0.192	0.095
	6–30	0.180	<0.0001	−1.888	<0.0001	0.059	0.466
	31–60			0		0	
Difficulty to not smoke (Ref: No)							
	Yes	0.094	0.0064	−7.395	<0.0001	0.030	0.749
Type of cigarette most like (Ref: others)							
	Morning	−0.020	0.5331	−4.896	<0.0001	0.048	0.589
More smoking in morning (Ref: No)							
	Yes	0.008	0.8282	3.846	<0.0001	0.194	0.021
Smoking when sick (Ref: No)							
	Yes	0.146	<0.0001	−0.093	0.571	0.156	0.061
Gender (Ref: Female)							
	Male	0.296	<0.0001	NA	NA	−0.128	0.354
Age group (Ref:50 + years)							
	–29	−0.197	<0.0001	1.842	<0.0001	−0.148	0.162
	30–49	−0.041	0.2353	−0.545	0.052	−0.04	0.586
Cotinine (ng/mL centered by median)							
		0.005	0.0004	0.125	<0.0001	0.006	0.042

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
