# Peer review of "Comparison of Nicotine Dependence and Biomarker Levels among Traditional Cigarette, Heat-Not-Burn Cigarette, and Liquid E-Cigarette Users: Results from the Think Study"

_ijerph, 2021, doi:10.3390/ijerph18094777_

Round 1

Reviewer 1 Report

I Would like to note that the urine collection from EC users was lower than
that of the other groups.

This group of EC users who collected urine is small and about 20 have low
dependence on the fagerstron score, therefore, conclusions about this group
are very limited, since of 377 users of EC, only 26 collected urine and
of these, 20 had Fagesrton below 5. 

I believe that this group has to be identified as to the daily liquid consumption.
This can be critical in relation to the conclusion.
The author considers that 10 puffs in the liquid is equal to 1 cigarette. 

The authors must inform this limitation in the analysis of the EC user group,
which makes the conclusion of the findings related to this subgroup very
limited. The TC and HNB groups have a larger N, therefore, more conclusive.

Author Response

I Would like to note that the urine collection from EC users was lower than that of the other groups.

This group of EC users who collected urine is small and about 20 have low dependence on the fagerstron score, therefore, conclusions about this group are very limited, since of 377 users of EC, only 26 collected urine and of these, 20 had Fagesrton below 5. I believe that this group has to be identified as to the daily liquid consumption.This can be critical in relation to the conclusion.

The author considers that 10 puffs in the liquid is equal to 1 cigarette. 

The authors must inform this limitation in the analysis of the EC user group,
which makes the conclusion of the findings related to this subgroup very
limited. The TC and HNB groups have a larger N, therefore, more conclusive.

Authors’ response:  Thank you. We also highly agree with reviewer’s comment that the interpretation of EC results is limited due to its small samples size and indirect method of cigarette consumption rate applied.

We already have had related sentences in our original version but we restated as seen below.

(Page 11, line 341)

However, due to the relatively small sample size of EC users, the comparison of biomarker levels was limited in interpreting the results of the study. And unlikely TC or HNBC, we considered that 10 puffs in the liquid is equal to 1 cigarette. It is necessary to conduct further analysis in the future for better validation purpose.

Reviewer 2 Report

General Comments: The purpose of this manuscript is to examine biomarker levels among different categories of tobacco users including those who use traditional cigarettes, e-cigarettes and heat not burn cigarettes.

Major points to consider:

  • While reviewing this manuscript, I noted the need for a more thorough definition and explanation of heat not burn cigarettes. This form of tobacco use is not widely available in the United States yet and, since the COVID-19 pandemic, I do not believe this option is widely known. I would request that the authors provide more detail on this option for those who are unfamiliar with this up and coming product.
  • I would like more detail on the smoking cessation questions asked and the purpose for asking these questions. Did you ask if the participants were using any nicotine replacement therapies for smoking cessation. In line 127-128, it seems as though these questions are asked as an after-thought and I would like some clarification and detail as to why these questions were included.
  • I believe the authors provide solid insight into their procedures. The organization and subheadings for this manuscript are clearly written.

Minor points to consider:

  • There are minor spelling and grammar errors found in lines 17, 57, 281, 354
  • Line 139: You may consider switching the order of this for clarity---instead of “non-smokers living with smokers or smokers” consider changing to “smokers and non-smokers living with smokers”.

I appreciate this manuscript and the clear comparison it offers. The tobacco use landscape is changes rapidly and this comparison between these categories of tobacco users certainly adds to the science.

Author Response

General Comments: The purpose of this manuscript is to examine biomarker levels among different categories of tobacco users including those who use traditional cigarettes, e-cigarettes and heat not burn cigarettes.

Major points to consider:

  • While reviewing this manuscript, I noted the need for a more thorough definition and explanation of heat not burn cigarettes. This form of tobacco use is not widely available in the United States yet and, since the COVID-19 pandemic, I do not believe this option is widely known. I would request that the authors provide more detail on this option for those who are unfamiliar with this up and coming product.

Authors’ response:  Thank you for this comment. The authors have explained in details the heat not burn cigarettes in the introduction.

HNBC which is a type of cigarette heated with an electric blade at 350°C, gives the smoker the true taste of tobacco, with reduced smoke and smell. The preference of smokers has been shifting from TC to EC and HNBC.

Since the introduction of IQOS, a type of HNBC, in South Korea in June 2017, such HNBCs have gained popularity, accounting for 10.5% of all total tobacco sales in 2019 [7, 8], and was advertised as a product with reduced harm [9].

  • I would like more detail on the smoking cessation questions asked and the purpose for asking these questions. Did you ask if the participants were using any nicotine replacement therapies for smoking cessation. In line 127-128, it seems as though these questions are asked as an after-thought and I would like some clarification and detail as to why these questions were included.

Authors’ response:  Thank you.

At the baseline, we excluded the users of these three products about various smoking cessation service programs (visiting smoking cessation clinics, using smoking cessation consultation calls, etc.).

  • I believe the authors provide solid insight into their procedures. The organization and subheadings for this manuscript are clearly written.

Authors’ response:  Thank you.

Minor points to consider:

  • There are minor spelling and grammar errors found in lines 17, 57, 281, 354

Authors’ response:  Thank you. We changed those typos : study, (line 17); them (line 57); Using the level of biomarkers among single-type cigarette smokers, (line 281); smoking (line 354).

  • Line 139: You may consider switching the order of this for clarity---instead of “non-smokers living with smokers or smokers” consider changing to “smokers and non-smokers living with smokers”.

Authors’ response:  Thank you. We changed it as seen below.

The survey of the 3004 participants (smokers or non-smokers living with smokers) included questions about age, gender, education, smoking habits

I appreciate this manuscript and the clear comparison it offers. The tobacco use landscape is changes rapidly and this comparison between these categories of tobacco users certainly adds to the science.

Round 2

Reviewer 1 Report

The author mention the limitation of
results regarding EC . The small sample size.